# Circulating P2X7 Receptor Signaling Components as Diagnostic Biomarkers for Temporal Lobe Epilepsy

**DOI:** 10.3390/cells10092444

**Published:** 2021-09-16

**Authors:** Giorgia Conte, Aida Menéndez-Méndez, Sebastian Bauer, Hany El-Naggar, Mariana Alves, Annette Nicke, Norman Delanty, Felix Rosenow, David C. Henshall, Tobias Engel

**Affiliations:** 1Department of Physiology and Medical Physics, Royal College of Surgeons in Ireland, University of Medicine and Health Sciences, D02 YN77 Dublin, Ireland; giorgiaconte@rcsi.com (G.C.); aidammendez@rcsi.ie (A.M.-M.); marianaalves@rcsi.ie (M.A.); dhenshall@rcsi.ie (D.C.H.); 2Epilepsy Center Hessen, Department of Neurology, Philipps-University Marburg, Baldingerstr, 35043 Marburg, Germany; S.Bauer@med.uni-frankfurt.de (S.B.); rosenow@med.uni-frankfurt.de (F.R.); 3Epilepsy Center Frankfurt Rhine-Main, Center of Neurology and Neurosurgery, Goethe-University Frankfurt, University Hospital Frankfurt, Schleusenweg 2-16 (Haus 95), 60528 Frankfurt am Main, Germany; 4LOEWE Center for Personalized Translational Epilepsy Research (CePTER), Goethe-University Frankfurt, Schleusenweg 2-16, 60528 Frankfurt am Main, Germany; 5Neurological Services, Beaumont Hospital, D09 V2N0 Dublin, Ireland; hanyelnaggar@rcsi.ie (H.E.-N.); normandelanty@beaumont.ie (N.D.); 6Walther Straub Institute of Pharmacology and Toxicology, Ludwig-Maximilians-Universität München, 80336 Munich, Germany; annette.nicke@lrz.uni-muenchen.de; 7School of Pharmacy and Biomolecular Sciences, Royal College of Surgeons in Ireland, University of Medicine and Health Sciences, D02 YN77 Dublin, Ireland; 8FutureNeuro, Science Foundation Ireland Research Centre for Chronic and Rare Neurological Diseases, Royal College of Surgeons in Ireland, University of Medicine and Health Sciences, D02 YN77 Dublin, Ireland

**Keywords:** status epilepticus, epilepsy, psychogenic non-epileptic seizures, diagnosis, biomarkers, inflammation, P2X7 receptor

## Abstract

Circulating molecules have potential as biomarkers to support the diagnosis of epilepsy and to assist with differential diagnosis, for example, in conditions resembling epilepsy, such as in psychogenic non-epileptic seizures (PNES). The P2X7 receptor (P2X7R) is an important regulator of inflammation and mounting evidence supports its activation in the brain during epilepsy. Whether the P2X7R or P2X7R-dependent signaling molecules can be used as biomarkers of epilepsy has not been reported. P2X7R levels were analyzed by quantitative ELISA using plasma samples from controls and patients with temporal lobe epilepsy (TLE) or PNES. Moreover, blood cell P2X7R expression and P2X7R-dependent cytokine signature was measured following status epilepticus in P2X7R-EGFP reporter, wildtype, and P2X7R-knockout mice. P2X7R plasma levels were higher in TLE patients when compared with controls and patients with PNES. Plasma levels of the broad inflammatory marker protein C-Reactive protein (CRP) were similar between the three groups. Using P2X7R-EGFP reporter mice, we identified monocytes as the main blood cell type expressing P2X7R after experimentally evoked seizures. Finally, cytokine array analysis in P2X7R-deficient mice identified KC/GRO as a potential P2X7R-dependent plasma biomarker following status epilepticus and during epilepsy. Our data suggest that P2X7R signaling components may be a promising subclass of circulating biomarkers to support the diagnosis of epilepsy.

## 1. Introduction

Epilepsy is a common chronic neurological disease, affecting up to 70 million people worldwide [1]. Temporal lobe epilepsy (TLE) is one of the most common and drug-refractory form of epilepsy in adults [1]. Together with the lack of effective pharmacological approaches in over 30% of patients and the absence of disease-modifying treatments, epilepsy diagnosis remains a clinical challenge, adding significantly to the disease burden [1,2]. While a correct diagnosis of epilepsy is important to inform treatment, to date, epilepsy diagnosis relies heavily on clinical examination, history, and patient monitoring via long-term video-encephalogram (vEEG) recording at hospitals. Long-term EEG recordings are, however, time-consuming, costly, have low throughput, and require a high level of specialist expertise [3]. Misdiagnosis rates are high and clinical signs can easily be confused with disorders, which present in a similar way, such as psychogenic non-epileptic seizures (PNES) [4]. Accordingly, there is significant interest in the discovery and validation of circulating (molecular) biomarkers to identify patients with epilepsy [5].

A role for neuroinflammation in epilepsy is well established [6]. Inflammatory mediators (e.g., Interleukin (IL)-1β, Tumor necrosis factor (TNF)-α) have been found increased in the brain following seizures and during epilepsy in both experimental models of epilepsy and patients [7]. Importantly, drugs targeting specific inflammatory pathways (e.g., Toll-like receptor 4 (TLR4), IL-1β, high mobility group box protein 1 (HMGB1)) provide potent anticonvulsive and antiepileptogenic effects [8,9,10]. Inflammatory mediators may also have diagnostic potential since levels of several inflammatory molecules have been found to be altered in the blood of patients with epilepsy [11]. This includes cytokines such as the pro-inflammatory cytokine IL-1β [12,13,14,15], HMGB1 [16], and the inflammation marker C-Reactive protein (CRP) [17].

The ATP-gated P2X7 membrane receptor (P2X7R) is a member of the ionotropic P2X receptor family that responds to extracellular-released adenosine triphosphate (ATP) [18,19]. P2X7Rs are highly expressed on cells of the immune system [20] both throughout the central nervous system (CNS), in particular on microglia [21], and on blood cells including lymphocytes, macrophages, and monocytes [22]. Due to its prominent role in driving inflammatory processes, the P2X7R has been implicated in numerous pathological conditions where inflammation is one of the underlying pathological hallmarks, including epilepsy [6,23]. P2X7R expression is increased in the brain following status epilepticus and during epilepsy in experimental models of epilepsy and patients [24,25,26,27,28]. Notably, P2X7R antagonism has been found to modulate seizure severity in acute seizure models including the intra-amygdala kainic acid (KA) mouse model of status epilepticus [23,25,26] and to reduce spontaneous seizure severity in epileptic rats [29] as well as spontaneous seizure frequency in epileptic mice [27].

While functional studies support a role for brain cell-expressed P2X7R in seizure generation and epilepsy, the presence of the P2X7R in the circulation in epilepsy has not been explored. Recent studies have shown P2X7R protein levels to be increased in the plasma of patients with underlying inflammation [30], including patients following sepsis [31]. Moreover, the expression of several cytokines downstream of P2X7R activation (e.g., IL-1β, IL-18) has been found altered in the blood of patients with epilepsy [12,13,14,15,32]. Whether P2X7R-dependent signaling can be used to support the diagnosis of epilepsy, has, however, not been investigated to date. Here, we assayed P2X7R levels in experimental and human TLE along with companion molecules, revealing a potential use of this protein as a circulating biomarker of epilepsy.

## 2. Materials and Methods

### 2.1. Patient Samples

Ethical approval was obtained from the medical research ethics committees at Beaumont Hospital, Dublin, Ireland (DUB/13/75) and from the local medical ethics committee at Marburg Hospital (Marburg, Germany) (MAR17/14). Written informed consent was obtained from all participants according to the Declaration of Helsinki principles. Plasma samples from healthy controls (N = 34; average age = 36 ± 2.14 years; males = 50%), TLE patients (N = 30; average age = 40.20 ± 2.96 years; males = 60%), patients with PNES (N = 11; average age = 30.1 ± 3.54 years; males = 18%), and patients with status epilepticus (N = 6; average age = 60.2 ± 3.89 years; males = 67%) were collected from the Epilepsy Monitoring Unit (EMU) at the Epilepsy Centre Hessen/Department of Neurology, Philipps University of Marburg, Germany and from the EMU at the Department of Neurology, Beaumont Hospital Dublin, Ireland. All patients admitted to the EMU had a detailed clinical assessment on admission including seizure types and frequency. All TLE patients were refractory to anti-seizure medication prior to admission and were on poly-drug therapy. Each patient at Beaumont was continuously video-EEG monitored using a standard international system electrode placement and computerized seizure detection was performed throughout the recording period. At Marburg Hospital, epilepsy was assessed by visual EEG assessment. The entire recording from the monitoring session was manually reviewed by a neurologist with special training in epilepsy. Blood samples were taken via venipuncture at baseline (following a seizure-free period of at least 24 h) and 1 h following a seizure. A detailed description of patient demographics can be found in Appendix A.

### 2.2. Human Plasma Preparation

Plasma samples were prepared as described previously [33]. Ensuring results are comparable between different hospital sites, protocols for plasma processing were harmonized between centers and followed a previously reported protocol based on specific guidelines. Ten mL of peripheral blood were collected from donors at baseline and 1 h post-seizure event using K2EDTA tubes (BD Bioscience, Franklin Lakes, NJ, USA). One hour post-blood collection, plasma was prepared by centrifuging the tubes at 1300× *g*, for 10 min, at 4 °C. Then, the supernatant was collected into an RNAase free tube and kept at −80 °C. Hemolysis in samples was assessed by spectrophotometric analysis using Nanodrop 2000 spectrophotometer. Absorbance at 414 nm was checked and samples with an absorbance > 0.25 were excluded from the study due to the possibility of hemolysis.

### 2.3. ELISA for the Human P2X7 Receptor

Human P2X7R protein levels in plasma were measured using a quantitative sandwich ELISA system kit (CUSABIO, Human P2X purinoceptor 7(P2RX7) ELISA kit (CSB-EL017325HU) Houston, USA) as reported previously [30]. The 96-well plates were previously coated by the manufacturer. The standard curve and plasma samples (100 μL) were added to the plate and incubated for 2 h at 37 °C. Then, plates were washed to eliminate excess and unspecific binding and incubated with the detection antibody (100 μL) for 2 h at 37 °C. Next, the second antibody, which is coupled to the substrate-modifying enzyme and binds to any antigen-antibody complexes, was added to the wells for 1 h at 37 °C. Finally, 100 µL of substrate, which is converted by the enzyme to elicit a chromogenic or fluorescent signal, was incubated for 15 min in the dark; 50 µL of Stop acid solution was added to each well and the absorbance was measured at 450 nm using 570 nm as reference. All buffers and solutions were included in the ELISA kit. To avoid batch effects between P2X7R plasma measurements or P2X7R-detecting ELISA plates used, results were normalized to the control group and are represented as percentage with controls set at 100%.

### 2.4. Animals

All animal experiments were performed in accordance with the principles of the European Communities Council Directive (2010/63/EU). Procedures were reviewed and approved by the Research Ethics Committee of the Royal College of Surgeons in Ireland (RCSI) (REC 1322) and Health Products Regulatory Authority (AE19127/P038; AE19127/P057). Mice used for our studies included 8–12-week-old male C57BL/6 OlaHsd wildtype mice, male C56BL/6N-P2rx7^tm1d(EUKOMM)wtsi^ P2X7 knock-out (KO, *P2X7^−/−^*) mice [34], and heterozygous male FVB/NJ mP2X7-EGFP BAC transgenic mice (FVB/N-Tg(RP24-114E20-P2X7/StrepHisEGFP)) which overexpress the P2X7 protein C-terminally fused to the enhanced green fluorescent protein (EGFP) expressed under the control of a BAC-derived *P2rx7* promoter (P2X7R-EGFP mice) [21]. For our studies, we used Line 17 from the initial generation of P2X7R-EGFP mice. This line showed the highest expression of P2X7R-EGFP and has been thoroughly characterized previously [21,28]. The original B6-P2rx7tm1a(EUCOMM)Wtsi strain was obtained from EMMA and crossed with the FLPe deleter mouse Gt(ROSA)26Sortm1(FLP1)Dym [35] to obtain the P2rx7fl/fl mouse, B6-P2rx7tm1c(EUCOMM)Wtsi. After removal of the FLPe, the P2rx7fl/fl mouse was crossed with the EIIa-Cre mouse, Tg(EIIa-cre)C5379Lmgd [36], to obtain, after removal of the EIIa-Cre, the *P2X7^−/−^* mouse. For further details see [21,37]. All mice were bred and housed in a controlled biomedical facility at RCSI, on a 12 h light/dark cycle at 22 ± 1 °C and humidity of 40–60% with food and water provided ad libitum.

### 2.5. Mouse Model of Status Epilepticus

Mice were anesthetized using isoflurane (5% induction, 1–2% maintenance) and maintained normothermic by means of a feedback-controlled heat blanket (Harvard Apparatus Ltd., Kent, UK). Fully anesthetized mice were placed in a stereotaxic frame and a midline scalp incision was performed to expose the skull. Then, a guide cannula (coordinates from Bregma; AP = −0.94 mm, L = −2.85 mm) for intra-amygdala injections was implanted and fixed in place with dental cement. Mice were removed from the frame and allowed to recover from anesthesia in a warmed incubator for a period of approximately 1 h. Status epilepticus was triggered via a microinjection of 0.3 µg KA (Sigma-Aldrich, Dublin, Ireland) in mice with a C57/Bl6 background (*P2X7^−/−^* mice and wild-type (*wt*) control) and 0.2 µg KA in mice with a FVB background (P2X7R-EGFP and *wt* littermates), diluted in 0.2 µL of phosphate-buffered saline (PBS) into the right basolateral amygdala, 3.75 mm below the dura, in immobilized (hand-restrained) awake mice. Non-status epilepticus control animals received an intra-amygdala microinjection of 0.2 µL of PBS. Seizures typically began within 5–10 min post-intra-amygdala KA injection and comprised long bursts of high-amplitude, high-frequency epileptiform activity. After 40 min post-intra-amygdala injection, all mice (KA and PBS-injected) received an intraperitoneal (i.p.) injection of lorazepam (6 mg/kg) (Wyetch, Taplow, UK) to reduce status epilepticus-induced morbidity and mortality. All mice subjected to intra-amygdala KA developed epilepsy after a short latent period of 3–5 days experiencing, 2–5 seizures per day [38,39,40]. Behavioral seizures were scored according to a modified Racine Scale as reported previously [26]. Score 1, immobility and freezing; Score 2, forelimb and or tail extension, rigid posture; Score 3, repetitive movements, head bobbing; Score 4, rearing and falling; Score 5, continuous rearing and falling; Score 6, severe tonic–clonic seizures. Mice were scored every 5 min for 40 min after KA injection. The highest score attained during each 5 min period was recorded by an observer blinded to treatment. Blood was collected into tubes containing 15 µL of 0.5 M EDTA (pH 7.4) via puncture of the saphenous vein either 1 h or 8 h after the administration of the anticonvulsant lorazepam, or once epilepsy was established (14 days post-KA injection). Note, mice were not video-EEG-monitored to confirm seizure frequency. Plasma was prepared as described for human samples. For measurement of cytokine concentrations in the hippocampi of mice, animals were killed via cervical dislocation 8 h post-status epilepticus, perfused with ice-cold PBS, and their brains were dissected and hippocampi were stored at −80 °C.

### 2.6. Cytokine Array (Hippocampus and Plasma)

Cytokine concentrations from *wt* and *P2X7^−/−^* mice (Control and 8 and 24 h post-intra-amygdala KA-induced status epilepticus) were measured in plasma and hippocampal homogenates using a multiplex electrochemiluminescence immunoassay kit (V-PLEX Plus Mouse Cytokine 19-Plex Kit, cat no K15255G, Meso Scale Discovery, Rockville, MD, USA) according to manufacturer instructions. Samples were prepared by homogenizing extracted hippocampi in ice-cold lysis buffer (20 mM pH 7.4, 100 mM NaCl, 20 mM NaF, 1% TritonTM X-100, 1 mM sodium orthovanadate, 1 M okadaic acid, 5 mM sodium pyrophosphate, 30 mM β-glycerophosphate, 5 mM EDTA, protease inhibitors) (Complete, Roche, Cat. No 11697498001). The antibody pre-coated plates allowed for the simultaneous quantification of the following cytokines: Interferon (IFN)-α, IL-1β, IL-2, IL-4, IL-5, IL-6, IL-9, IL-10, IL-33, IL-30, IL-12p70, IL-15, IL-17 A/F, Interferon gamma-induced protein (IP)-10, Keratinocyte chemoattractant (KC)/human growth-regulated oncogene (GRO), MCP-1, Macrophage Inflammatory Protein (MIP)-1α, MIP-2, and TNF-α. Samples (N = 2 for control (*wt* and *P2X7^−/−^*) and N = 4 for 8 h and 24 h post-status epilepticus (*wt* and *P2X7^−/−^*) were run in duplicates and plates were read with a Sector Imager 2400 (Meso Scale Discovery). Data were analyzed using the Discovery Workbench 4.0 software (Meso Scale Discovery).

### 2.7. ELISA for Cytokine and CRP Analysis

R&D system ELISA kits (DuoSet ELISA Ancillary Reagent Kit 2, DY008 Minneapolis, Minnesota, USA) were used to detect cytokines/chemokines, and human C-reactive protein levels (CRP) in plasma. The 96-well plate was coated overnight at room temperature (RT). After the coating, a standard curve and plasma samples were added to the plate with the following dilutions: 1:1000 for CRP, 1:5 for IL-1β, 1:10 for KC/GRO, and 1:2 for IL-6 and IL-18. The second antibody coupled with the substrate-modifying enzyme was added to the wells for 1 h at RT. Finally, 100 μL of substrate, which is converted by the enzyme to elicit a chromogenic or fluorescent signal, was incubated for 15 min in the dark. Fifty µL of Stop acid solution were added in each well and the absorbance was measured at 450 nm using 570 nm as reference. All buffers and solutions were included in the R&D system ELISA kits.

### 2.8. Fluorescence-Activated Cell Sorting (FACS)

Whole blood was collected from the mouse facial vein of transgenic P2X7R-EGFP mice (1 h post-lorazepam from vehicle-injected control mice and mice injected with intra-amygdala KA). Two-hundred µL of whole blood were mixed with 1 mL of red blood cell lysis buffer (Roche Diagnostic, Basel, Switzerland cat no11814389001) and shaken for 10 min. Samples were centrifuged at 500× *g* for 5 min and the supernatant was discarded. The cell pellet was resuspended in PBS for flow cytometry analysis. Flow cytometry analysis of blood cells was performed using AttuneTM NxT flow cytometer (ThermoFisher scientific, Waltham, MA, USA). Intact cells were identified by forward and side scatter [41]. GFP-positive cells were acquired excluding dead cells and doublets. Analysis of GFP-positive cells was performed using FlowJo v10 software, confirming gating in an unstained sample.

### 2.9. Statistical Analysis

Statistical analysis of data was performed using Prism 5 (GraphPad) and STATVIEW software (SAS Institute). Data are mean ± standard error of the mean (SEM). One-way ANOVA parametric statistics with post-hoc Fisher’s protected least significant difference test was used to determine statistical differences between three or more groups. Unpaired Student’s *t*-test was used for two-group comparison. Correlations between variables were assessed using Pearson’s correlation coefficient. Receiver Operating Characteristic (ROC) analysis was performed to investigate the diagnostic potential for P2X7R protein changes to identify TLE patients when compared with healthy controls and patients with PNES. Significance was accepted at *p* < 0.05.

## 3. Results

### 3.1. Increased P2X7R Plasma Levels in Patients with TLE

Plasma samples were obtained from healthy controls and patients admitted to two different epilepsy centers which included Marburg (Germany, MAR) and Dublin (Ireland, DUB) (Figure 1A and Appendix A). Plasma levels of the P2X7R in healthy controls was low but detectable at 190.4 ± 23.9 pg/mL (standard deviation), similar to previous reports [30]. P2X7R plasma levels were higher in baseline samples from patients with TLE (242.6 ± 39.2 pg/mL) compared with controls and levels remained similarly elevated in samples collected 1 h following an EEG-detected seizure (302.1 ± 86.3 pg/mL) (Figure 1B). P2X7R plasma levels were also significantly raised in TLE patients compared with patients diagnosed with PNES (161.8 ± 31.8 pg/mL). P2X7R levels were similar between patients with PNES and controls (Figure 1B). While increased by approximately 20% in patients suffering from status epilepticus (287.5 ± 79.7 pg/mL), when compared to control, this did not reach statistical significance (*p* = 0.0831), which was probably due to the low N number of our status epilepticus patient cohort. P2X7R protein levels were, however, higher in patients with SE when compared with patients with PNES (Figure 1B). Importantly, when samples were analyzed separately for each hospital, similar trends between patient groups were observed with TLE patients showing higher P2X7R protein levels in plasma when compared with control and PNES patients, ruling out possible batch effects between the two analyzed cohorts (Appendix A). ROC analysis demonstrated that measurements of P2X7R plasma levels could differentiate between controls and TLE patients with a moderate level of sensitivity (60%) and good specificity (74%) with an area under the curve (AUC) of 0.64 (Figure 1C). Likewise, ROC analysis comparing TLE patients with patients with PNES showed a high sensitivity (90%) and specificity (63%) with an AUC of 0.77 (Figure 1D). Interestingly, P2X7R plasma levels were higher in 7 out of 10 samples taken 1 h post-seizure when compared with the available corresponding baseline sample (Figure 1E), suggesting seizures further increase plasma P2X7R levels in TLE patients. Analysis of potential co-variables found no significant differences in P2X7R plasma levels in controls and TLE patients according to gender (Control: *p* = 0.33; TLE: *p* = 0.92) (Figure 1F) or according to age (Control: r^2^ = 0.03, *p* = 0.33; TLE: r^2^ = 0.003, *p* = 0.72) (Figure 1G). Moreover, no obvious effects of different ASDs on P2X7R plasma levels were observed and no significant correlation was found between P2X7R plasma levels and seizure frequency (r^2^ = 0.05; *p* = 0.34) and duration of epilepsy (r^2^ = 0.18, *p* = 0.08) (Appendix A). Finally, there is evidence that expression of the P2X7R may vary according to time of day [42]. Accordingly, we compared P2X7R levels in a subset of TLE patients. Analysis of plasma levels of P2X7R from TLE patient samples taken in the morning versus afternoon showed no difference (N = 8 per time-point (TLE baseline), *p* = 0.604).

### 3.2. Correlation between P2X7R Plasma Levels and the Inflammation Marker CRP in TLE Patients

Because the P2X7R has been shown to be an important driver of inflammation [20] and because peripheral inflammation is one of the hallmarks of epilepsy [43], we next sought to establish whether P2X7R plasma levels correlate with the inflammation marker CRP. CRP levels were measured in a subset of samples from the same healthy controls and patients used to quantify P2X7R levels. This revealed that, in contrast to P2X7R plasma levels, CRP plasma levels were similar between TLE patients when compared to healthy controls and patient with PNES (Controls (100.0 ± 7.19, N = 13) vs. TLE baseline (107.0 ± 9.78, N = 26), *p* = 0.60) (Figure 2A). CRP plasma levels seemed to be, however, slightly increased in patients with status epilepticus when compared with control. This did, however, not reach significance, which was probably due to the low patient number for this group (*p* = 0.12) (Figure 2A). Next, we investigated a putative correlation between CRP and P2X7R plasma levels. Linear regression analysis demonstrated a strong positive correlation between plasma P2X7R and CRP levels 1 h post-seizure (r^2^ = 0.42, *p* = 0.008) (Figure 2B). A similar relationship was evident, although not significant, when comparing these proteins in samples from patients who experienced status epilepticus (r^2^ = 0.49, *p* = 0.11) (Figure 2C). P2X7R and CRP levels showed no correlation in baseline epilepsy samples (r^2^ = 0.0002, *p* = 0.94) or in healthy controls (r^2^ = 0.0029, *p* = 0.86) (Figure 2D,E).

### 3.3. P2X7 Receptor Expression Increases in Blood Cells following Status Epilepticus in Mice

To extend these insights and to explore the cell type(s) that express the P2X7R in epilepsy, we subjected mice expressing a C-terminally EGFP-tagged form of the P2X7R to status epilepticus [38]. KA-induced seizures were curtailed by lorazepam and blood was collected 1 h later followed by measurement of GFP-positive blood cells via flow cytometry (Figure 3A). The number of EGFP-positive cells was significantly increased in samples from mice subjected to status epilepticus (Figure 3B). Moreover, a P2X7R-EGFP-positive cell population in white blood cells in this transgenic model was clearly visible, with the distribution of P2X7R-EGFP in the forward-side scatter graph suggesting these to be likely composed of monocytes [41,44] (Figure 3C).

### 3.4. Increased KC/GRO Plasma Levels in P2X7^−/−^ Mice Post-Status Epilepticus

The P2X7R is an important regulator of the release of cytokines in the brain and circulation [20,45], suggesting changes in circulating cytokines could be potential readouts or surrogates of P2X7R activity/inhibition. Moreover, circulating cytokines have been proposed as possible diagnostic tools for numerous brain diseases, including epilepsy [46]. To explore whether the P2X7R contributes to a specific cytokine signature following seizures, we measured cytokine levels in brain tissue (hippocampus) and plasma under control conditions and 8 and 24 h following status epilepticus in *wt* mice and mice deficient in P2X7R (*P2X7^−/−^*) using the V-PLEX Plus Mouse Cytokine 19-Plex Kit (Figure 4A). This assay measures levels of nineteen different cytokines and chemokines (see Materials and Methods and Appendix A).

Both genotypes experienced similar seizures during status epilepticus (Figure 4B), in line with previous studies using the same transgenic *P2X7^−/−^* mouse model [34]. This suggests differences in cytokine profiles are not due to differences in seizure severity between genotypes. Cytokine profiles in *wt* mice changed in the hippocampus, as expected, following status epilepticus, including elevations in IL-1β, TNF-α, and MCP-1 (Appendix A and Appendix A) [43,47]. Unexpectedly, levels of several cytokines were increased in mice lacking the P2X7R compared with *wt* after status epilepticus including both anti-inflammatory cytokines (e.g., IL-10, IL-2, IL-4) and pro-inflammatory cytokines (e.g., IL-6 and TNF-α) and chemokines (e.g., MIP-1α, MIP-2, KC/GRO) (Appendix A). This indicates that the P2X7R may suppress as well as promote inflammatory signaling and that factors in addition to the P2X7R contribute to the release of cytokines.

Based on these findings, we selected cytokines to validate the array-based screen in a separate cohort of mice via ELISA. This included the known P2X7R targets IL-1β and IL-6 [45] and the cytokine KC/GRO which, although not previously described to be regulated by the P2X7R, appeared to be differently regulated between genotypes in both the hippocampus and plasma according to our array results (Appendix A and Appendix A). In addition, we also included IL-18, which is a well-known down-stream signaling molecule of P2X7R [20] that was, however, not included in the array. Plasma cytokine levels were individually analyzed 8 h post-status epilepticus and 14 days post-status epilepticus, a time-point when mice subjected to intra-amygdala KA normally experience epileptic seizures [38,39,40]. KC/GRO levels were increased in *P2X7^−/−^* mice in plasma following status epilepticus and epilepsy (Figure 4D), in good agreement with our array results (Appendix A and Appendix A). No changes could be observed in plasma levels between genotypes for IL-1β (Figure 4E) and IL-6 (Figure 4F) post-status epilepticus and during epilepsy. While the cytokine array revealed similar IL-1β plasma levels for *wt* and *P2X7^−/−^* mice, IL-6, in contrast to our ELISA results, showed a slight increase in *P2X7^−/−^* mouse plasma 24 h post-status epilepticus (Appendix A and Appendix A). Finally, no significant changes in IL-18 plasma levels were observed following status epilepticus or during epilepsy between genotypes (Figure 4G). Taken together, these studies suggest the P2X7R may negatively regulate plasma levels of the cytokine KC/GRO, and that this cytokine may therefore represent a potential peripheral biomarker of P2X7R activation.

## 4. Discussion

Here, we report that P2X7R protein levels were increased in the plasma of patients with TLE when compared with healthy controls and patients with PNES. Our study further identified KC/GRO as a potential P2X7R-dependent cytokine in plasma post-status epilepticus and during epilepsy. Thus, ELISA-based detection of P2X7R and P2X7R downstream molecules could offer a promising biomarker for epilepsy and a method upon which to build diagnostic tools.

While P2X7R expression in the brain has repeatedly been shown to increase following status epilepticus and during epilepsy [23,48], whether P2X7R expression also changes in blood during epilepsy has, to our knowledge, not been investigated to date. Here, we reported increased P2X7R protein plasma levels in TLE patients (baseline and following seizures). P2X7R levels were also increased in patients following an episode of status epilepticus. There remains an unmet need for biomarkers to support the diagnosis of epilepsy and to differentiate between patients with epilepsy from alternative presentations, such as PNES [4]. Several classes of potential biomarkers for seizures and epilepsy are currently under investigation. These include imaging and electrophysiological-based methods, genetic markers, and changes in gene expression and metabolite concentrations in tissues with circulating biomarkers attracting particular attention. Among these, circulating non-coding RNAs such as microRNAs and transfer RNA fragments have proven particularly promising candidates [5,11,33,46,49]. However, possible disadvantages of these biomarkers include the lack of a rapid and cost-efficient analysis platform or the need for difficult-to-access biofluids (e.g., cerebrospinal fluid (CSF)), the requirement of large volumes of blood, or the need for further blood processing, possibly contributing to inter-hospital variability [50]. Inflammation markers are another class of biomarkers increasingly recognized as promising diagnostic tools for both seizures and epilepsy [11]. This includes different cytokines, several members of the complement cascade, markers of neuronal injury (e.g., Neuron-specific enolase (NSE)) and astroglial response (e.g., Glial fibrillary acidic protein (GFAP), protein S100β). The pro-inflammatory signaling molecule HMGB1 has attracted particular attention as a possible mechanistic biomarker for epilepsy, with studies showing that, similar to the P2X7R, drugs interfering with HMGB1 signaling provide potent anticonvulsive and antiepileptic effects [51]. Similar to other biomarkers, the detection of circulating P2X7R requires the processing of blood, possibly leading to differences between studies. Our analysis was, however, carried out at two different hospitals obtaining comparable results. Moreover, previous published data have shown P2X7R plasma levels to be stable regardless of storage or fluid used (plasma vs. serum) [30]. Another possible advantage is the detection of P2X7R via ELISA, which allows for a quantitative measurement of P2X7Rs, critical for patient stratification. While we recognize that a 44% increase in P2X7R plasma levels in TLE patients when compared to control may be challenging to use as a stand-alone measure to diagnose epilepsy, the differences in P2X7R plasma levels were >100% when comparing TLE patients with PNES patients. This is of particular relevance, as it is critical to distinguish patients experiencing real epileptic seizures from patients affected by psychogenic ones. As mentioned before, misdiagnosis between these conditions remains high and may lead to unnecessary treatments with the associated ASD-induced side effects and possible delays in giving psychological therapy to PNES patients [4,52]. Of note, all patients enrolled in our study were diagnosed via gold-standard video EEG including patients with PNES.

The causes of increased P2X7R plasma levels in TLE patients remains to be determined. P2X7R activation is heavily linked to inflammatory processes and P2X7R expression has been shown to be increased in the blood in pathologies with an underlying inflammatory condition such as infections, sepsis, and diabetes [30,31,53]. An unanticipated finding was, therefore, that CPR levels were similar between TLE patients and controls in our study. However, while a significant association between CRP levels and epilepsy has been observed in some studies [54,55], this was not confirmed by others in line with our results [17,56]. The reasons for these discrepancies remain to be determined, however differences in underlying pathologies and etiologies may explain some of these. This is an important issue; consequently, future studies must be designed to establish why some patients show altered CRP levels while others do not. Notably, our data suggested a better efficiency of P2X7R plasma levels for the diagnosis of epilepsy when compared with the inflammation marker CRP. However, while we found that P2X7R plasma levels strongly correlated with the inflammatory marker CRP shortly following seizures, no correlation was found during baseline conditions in TLE patients or in healthy controls. Of note, despite the strong correlation of P2X7R and CRP plasma levels 1 h post-seizures, only P2X7R plasma levels, and not CRP plasma levels, were higher when compared with control levels. This is unexpected, as one would expect that both P2X7R and CRP would show the same increases in relation to controls. However, CRP plasma concentrations were only measured in a subset of controls and patients and both CRP and P2X7R plasma levels showed a high inter-sample variability, which may explain why P2X7R levels but not CRP levels, despite showing a positive correlation 1 h post-seizures, were increased when compared with control. Nevertheless, our data suggested that while inflammatory processes may contribute to increases in P2X7R plasma expression during seizures, other inflammation-independent processes may also contribute during epilepsy.

The source of increased P2X7R levels detected in plasma remains uncertain. Previous work has shown P2X7Rs to be released from monocytes in blood [30]. In line with P2X7R stemming from blood cells, our results using P2X7R-EGFP reporter mice show increased P2X7R-EGFP expression on blood cells 1 h post-status epilepticus. Interestingly, as suggested from the forward side scatter [41,44], this increase seems to be mainly restricted to monocytes, main blood cells expressing P2X7Rs [53,57,58]. Other sources are, however, also possible, including the brain. Among brain cells, microglia are probably the most likely source of increased P2X7R levels in plasma. Microglia are one of the first cell types to respond to injury [59] and P2X7Rs are highly expressed on microglia following status epilepticus and during epilepsy [28]. While not detected on astrocytes, P2X7Rs have, however, also been shown to be increased on neurons and oligodendrocytes during epilepsy [25,28], making these cell types also possible sources. Therefore, future studies should be carried out to determine the exact source of P2X7R protein during epilepsy.

While promising, we acknowledge, however, that changes in P2X7R protein levels in plasma are not unique to seizures and epilepsy and, as mentioned above, have been previously reported for different pathological conditions [30,31,53,60]. Nevertheless, single biomarkers are unlikely to be used as a stand-alone test and will be evaluated within a clinical context in combination with other measures. Our results should, however, be further validated in a multi-center study with a larger patient cohort including non-TLE epilepsy patients and patients with status epilepticus to allow for a better understanding of the relationship between plasma P2X7R levels and epilepsy syndromes, underlying disease phenotype, and preictal, ictal, postictal, and interictal states. Moreover, P2X7R expression has been reported to change according to the time of day [42]. Future studies should, therefore, investigate whether P2X7R expression in plasma undergoes the same changes.

The second major finding of our study was the identification of the cytokine KC/GRO as a potential P2X7R down-stream target post-status epilepticus and during epilepsy with KC/GRO levels found elevated in plasma of *P2X7^−/−^* mice. KC/GRO levels were, however, not significantly increased during control conditions, suggesting that the blocking of the release of KC/GRO via P2X7R is restricted to pathological conditions (i.e., status epilepticus, epilepsy) when high amounts of extracellular ATP are available to activate the P2X7R [61]. In contrast to our studies which showed no differences in KC/GRO levels between control conditions and status epilepticus/epilepsy in plasma, previous data have shown increased KC/GRO protein levels in the hippocampus of rats 12 h post-status epilepticus [62], in the brain and in plasma of rats with non-convulsive status epilepticus [63], and in the CSF of pediatric patients with febrile infection-related epilepsy syndrome [64]. Differences in models used, timing of sampling post-status epilepticus, or tissue may account for these observed differences between studies, which should be addressed in the future. Nevertheless, to our knowledge, there are no previous reports linking this chemokine directly with the P2X7R. KC/GRO (also known as CXCL1) together with MIP-2 (also known as CXCL2) are the two major chemoattractants responsible for the recruitment of neutrophils and both bind to the chemokine receptor, CXCR2 [65]. Interestingly, studies blocking the P2X7R have shown decreased neutrophil infiltration into the brain post-status epilepticus [66]. Moreover, these chemokines have been shown to be involved in the release of IL-1β via activating the NLR family pyrin domain containing 3 (NLRP3) inflammasome and KC/GRO-MIP2-mediated regulation of the NLRP3 inflammasome, which seems to be independent from the NLRP3 activation via P2X7R [67]. Thus, KC/GRO overexpression may compensate for the absence of the P2X7R, thereby possibly maintaining the activation of the NLRP3 inflammasome and the levels of IL-1β unchanged in the blood.

An unexpected finding in the present study was that knock-out of the P2X7R was associated with higher release of cytokines post-status epilepticus, including both pro- and anti-inflammatory cytokines (e.g., IL-10, IL-12p70, IL-27/30, IL-4, KC/GRO MIP-1α, MIP-2, TNF-α). This indicates that the P2X7R may negatively regulate some inflammatory pathways in addition to its role in promoting release of IL-1β and other cytokines/chemokines [68,69,70,71,72]. In particular, the lack of differences in IL-1β plasma levels between genotypes was unexpected as previous data has shown decreased IL-1β expression in peritoneal macrophages in *P2X7^−/−^* mice when compared with *wt* mice following a stimulation with ATP [73]. Likewise, no changes in plasma levels were observed for IL-18 nor IL-6, previously associated with the P2X7R [45]. There are several potential explanations for these differences, including the use of different transgenic *P2X7^−/−^* mice (e.g., *P2X7^−/−^* mice previously used to identify P2X7R-dependet cytokines in the blood [73], expressing several P2X7R splice variants within the CNS [74]), different disease backgrounds, or models used (e.g., cells vs. whole organism). Other cytokine release-regulating proteins, including other P2 receptors, may become activated during seizures/epilepsy, masking the effects of the P2X7R. P2Y_1_R, P2Y_12_R, and P2X4R, all been shown to be activated during seizures [48] and to regulate the release of cytokines [75,76]. Compensatory effects in *P2X7^−/−^* mice are also a possible reason. Differences in seizure severity and epilepsy development between genotypes may also compensate changes in cytokine expression or increase differences between genotypes. However, as reported previously [34], both *wt* and *P2X7^−/−^* mice experienced similar seizures during status epilepticus. Whether this is the case for epilepsy development remains to be established. One potential concern with the use of the current *P2X7^−/−^* mouse model is the inclusion of exon 1. This could potentially leave functional P2X in lymphocytes. However, P2X7Rs depend on the presence of both transmembrane domains (TMs) to form functional receptors. Since TM1 is partly encoded by exon 2, deletion of this exon should result in the knock-out of all functional rodent splice variants, including P2X7a and P2X7k.

Finally, similar to KC/GRO levels, no changes were observed between conditions (control vs. status epilepticus/epilepsy) in plasma concentrations for IL-1β, IL-6, or IL-18 in *wt* mice. In line with these results, IL-1β expression in plasma was unaltered in patients with epilepsy at different time-points following seizures [13,77]. Regarding IL-18, conflicting results have been reported. While increased IL-18 protein levels have been observed in serum of epileptic patients compared with healthy controls [32], another study found a slight decrease of IL-18 in mesial TLE and focal epilepsy patients [78]. With regards to IL-6, previous studies in epilepsy patients showed that IL-6 levels are higher acutely post-seizures [77], while the baseline level of IL-6 in chronic epilepsy are lower [13]. Thus, while the potential use of cytokines as diagnostics for seizures and epilepsy requires further, more detailed investigation, circulating cytokines may be a useful tool as read-out of P2X7R activation status.

## 5. Conclusions

In summary, our data offered the proof-of-concept that the detection of P2X7R in plasma is a promising novel diagnostic tool capable of supporting a differential diagnosis of epilepsy.

## Figures and Tables

**Figure 1 cells-10-02444-f001:**
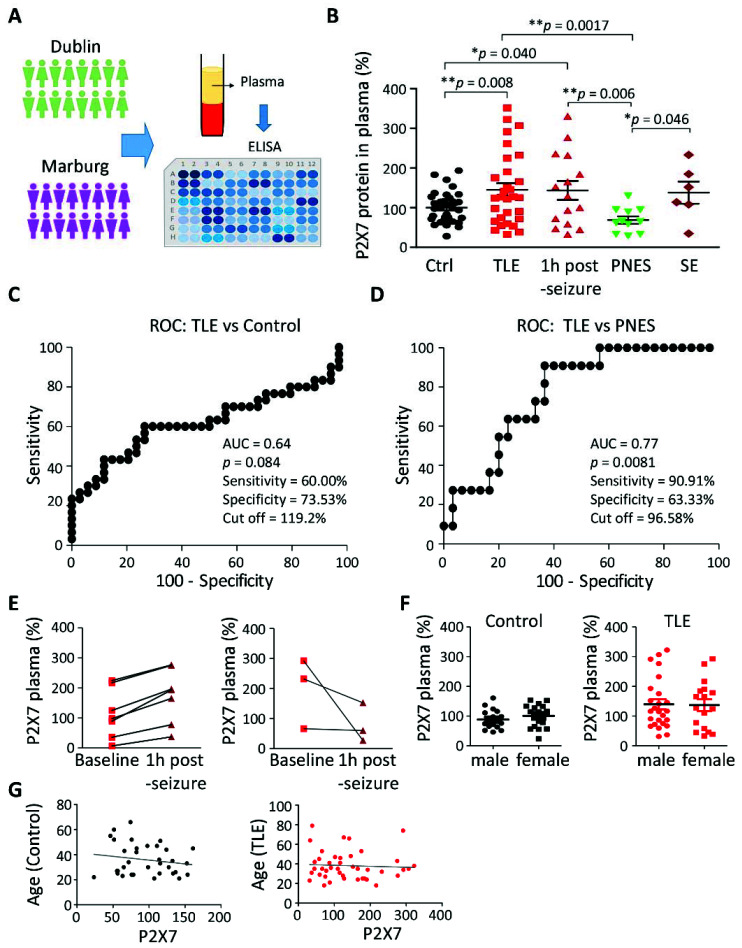
Increased P2X7R protein levels in plasma of patients with TLE. (**A**) Patients were recruited from two hospitals: Marburg Hospital (Germany) and Beaumont Hospital (Ireland), and plasma samples were analyzed via P2X7R-detecting ELISA. (**B**) Bar chart showing P2X7R protein levels in plasma from temporal lobe epilepsy (TLE) patients (N = 30) in baseline conditions and 1 h post-seizure (N = 15) are higher when compared with healthy controls (Ctrl) (N = 34) and patients with psychogenic non-epileptic seizures (PNES) (N = 11). In addition, P2X7R protein levels were also higher in patients suffering from an episode of status epilepticus (SE) (N = 6) when compared with PNES patients. ANOVA with post-hoc Fisher correction. Data are given as percentage to control. (**C**) ROC curve analysis shows P2X7R plasma levels have a moderate sensitivity (60%) and good specificity (74%) for discriminating between healthy controls and TLE patients with an AUC of 0.64 at a cut-off of 119.2%. (**D**) ROC curve analysis demonstrating a good sensitivity (90%) and specificity (63%) for discriminating between TLE patients and patients with PNES with an AUC of 0.77 at a cut-off of 96.58%. (**E**) Dot plot showing P2X7R plasma levels in the same patients at baseline and 1 h post-seizure. Out of 10 patients, 7 showed increased P2X7R in plasma and 3 lower P2X7R plasma levels 1 h post-seizures when compared with the corresponding baseline levels. (**F**,**G**) No correlation of P2X7R plasma levels according to sex (F) (Control: *p* = 0.33; TLE: *p* = 0.92) or age (**G**) (Control: r^2^ = 0.03 *p* = 0.33; TLE: r^2^ = 0.003, *p* = 0.72). * *p* < 0.05, ** *p* < 0.01.

**Figure 2 cells-10-02444-f002:**
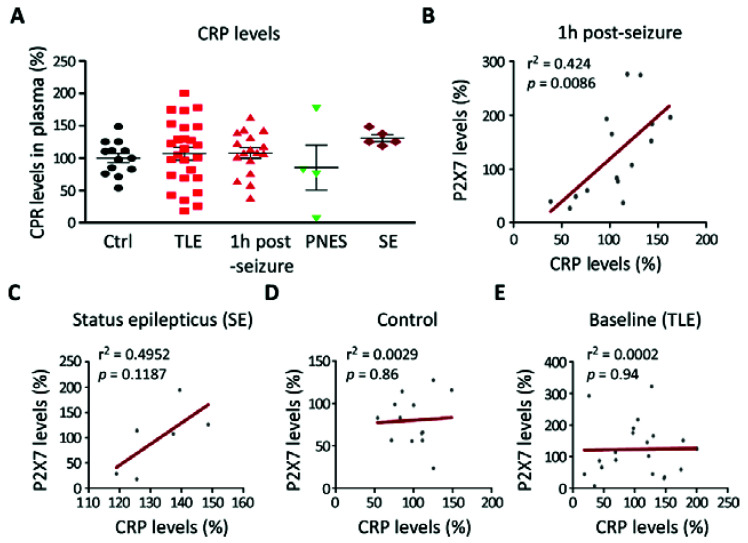
Correlation between P2X7R plasma levels with CRP plasma levels. (**A**) Bar chart showing C-Reactive protein (CRP) plasma levels in healthy controls (Ctrl) (N = 13), TLE patients (N = 26), 1 h post-seizure (N = 17), patients with PNES (N = 4), and patients suffering from status epilepticus (SE) (N = 5). Data are given as percentage to control. (**B**–**E**) Correlation between CRP plasma levels and P2X7R plasma levels 1 h post-seizure (N = 15; r^2^ = 0.42, *p* = 0.0086), during status epilepticus (N = 6; r^2^ = 0.49, *p* = 0.11), in healthy controls (N = 13, r^2^ = 0.0029, *p* = 0.86), and in TLE patients at baseline (N = 19; r^2^ = 0.0002, *p* = 0.94).

**Figure 3 cells-10-02444-f003:**
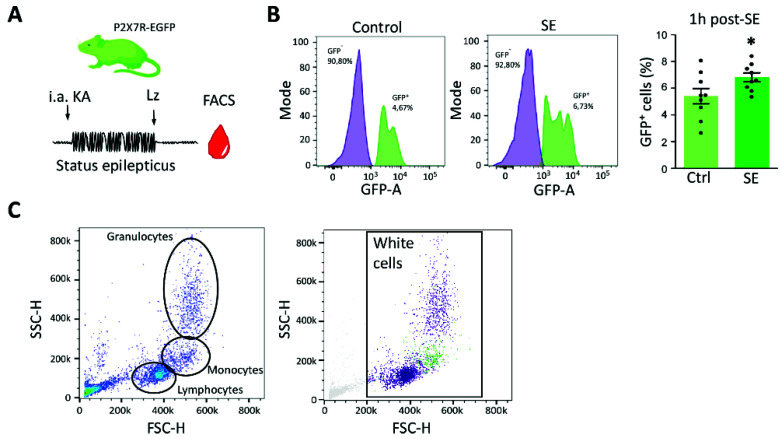
Increased P2X7R expression in blood cells post-status epilepticus in mice. (**A**) Mice expressing P2X7 C-terminally fused to EGFP are subjected to intra-amygdala KA-induced status epilepticus. Blood was collected 1 h post-treatment with the anticonvulsant lorazepam. (**B**) Graph showing the histogram of the GFP-A (area) in control conditions and 1 h post-SE. The percentage of GFP^+^ cells is significantly increased in animals treated with intra-amygdala KA 1 h post-lorazepam (SE) when compared with vehicle-injected control mice (Ctrl) (N = 9 per group) (Unpaired Student’s *t*-test, *p* = 0.0493). (**C**) Forward side scatter indicating the distribution of the GFP^+^ cell population (in green). Note, size and complexity suggests the GFP^+^ cell population being composed mainly of monocytes. * *p* < 0.05.

**Figure 4 cells-10-02444-f004:**
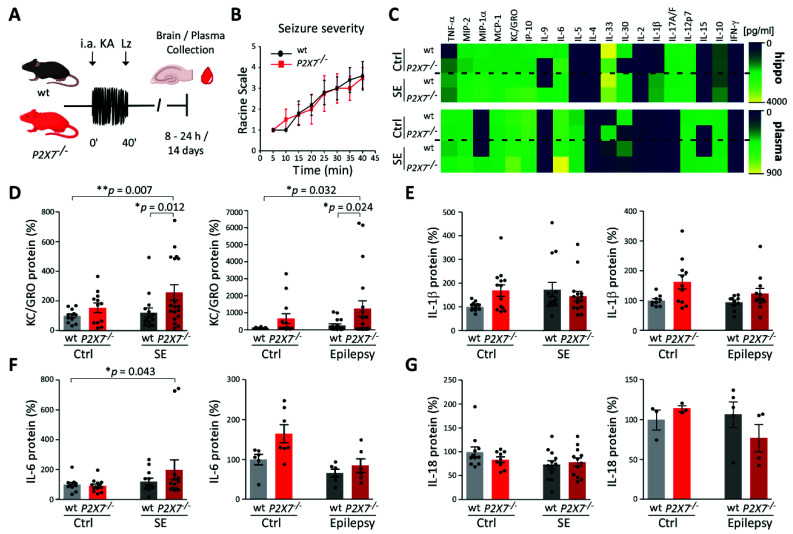
Increased KC/GRO expression in *P2X7^−/−^* mice post-status epilepticus and during epilepsy. (**A**) Experimental design. *Wt* and *P2X7^−/−^* mice were subjected to intra-amygdala KA-induced status epilepticus. Hippocampi were collected 8 h and 24 h post-lorazepam treatment. Blood samples were taken 8 h, 24 h, and 14 days post-KA. (**B**) Graph showing Racine scores (maximum reached) per 5 min period over 40 min after injection of KA for *wt* and *P2X7^−/−^* mice (n = 5 (*wt*) and 4 (*P2X7^−/−^*). (**C**) Heat map showing changes in cytokine concentrations in *wt* and *P2X7^−/−^* mice in control conditions in plasma and hippocampus 8 h following status epilepticus as measured via cytokine array. Blue color represents lower cytokine concentrations and yellow color higher cytokine concentrations. (**D**) KC/GRO levels in plasma of *wt* and *P2X7^−/−^* mice under control conditions and 8 h post-status epilepticus (SE) (control: N = 12 per genotype; post-SE: N = 15 (*wt*) and 18 (*P2rx7^−/−^*)) and during epilepsy (14 days post-KA injection) (control: N = 12 (*wt*) and 13 (*P2rx7^−/−^*); epilepsy: N = 16 (*wt*) and 19 (*P2X7^−/−^*)). (**E**) IL-1β levels in plasma of *wt* and *P2X7^−/−^* mice under control conditions and 8 h post-status epilepticus (control: N = 12 (*wt*) and 13 (*P2X7^−/−^*); post-SE: N = 14 (*wt*) and 15 (*P2X7^−/−^*)) and during epilepsy (control N = 9 (*wt*) and 11 (*P2X7^−/−^*); epilepsy: N = 11 (*wt*) and 12 (*P2X7^−/−^*)). (**F**) IL-6 levels in plasma of *wt* and *P2X7^−/−^* mice under control conditions and 8 h post-status epilepticus (control: N = 12 (*wt*) and 13 (*P2X7^−/−^*); post-SE: N = 12 (*wt*) and 13 (*P2X7^−/−^*)) and during epilepsy (control: N = 6 (*wt*) and 7 (*P2rx7^−/−^*); epilepsy: N = 6 per genotype). (**G**) IL-18 levels in plasma of *wt* and *P2X7^−/−^* mice under control conditions and 8 h post-status epilepticus (control: N = 11 (*wt*) and 9 (*P2rx7^−/−^*); post-SE: N = 12 per genotype) and during epilepsy (control: N = 4 per genotype; epilepsy: N = 4 per genotype) (D–H). Data are represented as percentage to *wt* control. ANOVA with post-hoc Fisher correction test. * *p* < 0.05, ** *p* < 0.01.

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
