# Peer review of "Circulating P2X7 Receptor Signaling Components as Diagnostic Biomarkers for Temporal Lobe Epilepsy"

_cells, 2021, doi:10.3390/cells10092444_

Round 1

Reviewer 1 Report

The authors of this study provide evidence that there is an increase in P2X7R protein levels in plasma from patients with temporal lobe epilepsy (TLE), under basal conditions and following seizures, as well as in patients following an episode of status epilepticus, but not in patients with psychogenic non-epileptic seizures (PNES). This is an new observations because an increase in P2X7R expression during epilepsy was previously found only in the brain samples. They also show that P2X7R plasma level is a better marker for the diagnosis of epilepsy than the broad inflammation marker C-Reactive protein (CRP). Comparable results have been obtained by analysis of blood samples that was carried out at two different hospitals (in Germany and Ireland), further supporting the conclusion that circulation P2X7R protein levels might be used as potential biomarker for the diagnosis of epilepsy.

In animal experiments the authors used mice expressing a C-terminally EGFP-tagged form of the P2X7R. Measurement of GFP-positive blood cells revealed that the number of EGFP-positive cells was significantly increased in samples from mice subjected to status epilepticus by the intra-amygdala kainic acid injection, and that a P2X7R-EGFP-positive cells were likely monocytes. They also found that levels of several cytokines, including cytokine KC/GRO, were increased in mice lacking P2X7R compared to wild type after status epilepticus, indicating that also other factors in addition to the P2X7R contribute to the release of cytokines and that plasma levels of the cytokine KC/GRO may also represent a potential biomarker for diagnosis of epilepsy.

Major Comments:

A role for neuroinflammation in epilepsy is well established, but this study shows that plasma levels of the broad inflammatory marker protein CRP were similar in TLE patients when compared to controls (Fig.2A). On the other hand, the authors found that P2X7R plasma levels were higher in TLE patients when compared to controls, and that there is a “strong positive correlation between plasma P2X7R and CRP levels 1 h post-seizure“ (line 298) and in status epilepticus (Fig. 2B and 2C). This discrepancy is explained nowhere in the text.

Minor Comments:

Line 29: Abbreviations „TLE” and “PNES” should be explained also in Abstract

Line 109: Since at the mRNA level, P2X7 shows a time of day dependent variation (Lommen  et al, 2017, Cell and Tissue Research 369:579-590), it should be specified at what time exactly the samples were collected.

Line 246 - 248: “P2X7R plasma levels were higher in baseline samples from patients with TLE (242.6 ± 39.2 pg/ml) compared to controls and levels remained similarly elevated in samples collected 1 h following an EEG-detected seizure (302.1 ± 86.3 pg/ml) (Figure 1B)“ But Fig. 1B shows statistical comparison of SE with PNES, not with control, as mentioned in text to this figure: “P2X7R protein levels were also higher in patients suffering from an episode of status epilepticus (SE) (N = 6) when compared to PNES patients” Was the difference between control and TLE statistically significant?  

Line 308: Correlation

Line 327: Number of mice in SE group should be also given

Lines 450-451: “P2X7R levels seemed to be even below control levels in patients with PNES..“. This differnce was not mentioned to be significant in Results. Moreover, (line 245) the authors stated that " P2X7R levels were similar between patients with PNES and controls (Figure 1B)"

Lines 458 -459: “Other sources are, however, also possible including the brain.” This sentence is vague. The authors should specify cells in the brain which could be considred as a P2X7R source - microglia, ependymal cells or astrocytes?

Lines 502 – 511:” Potential explanations for these differences….“. This part of Discussion is full of speculations and poor arguments. For example, what type of P2 receptors, except P2X7, could facilitate release of cytokines? How different could be „different transgenic P2X7-/- mice“?

Author Response

  1. A role for neuroinflammation in epilepsy is well established, but this study shows that plasma levels of the broad inflammatory marker protein CRP were similar in TLE patients when compared to controls (Fig.2A). On the other hand, the authors found that P2X7R plasma levels were higher in TLE patients when compared to controls, and that there is a “strong positive correlation between plasma P2X7R and CRP levels 1 h post-seizure“ (line 298) and in status epilepticus (Fig. 2B and 2C). This discrepancy is explained nowhere in the text.

We strongly agree with the reviewer and would like to thank the reviewer for pointing this out. Indeed, peripheral inflammation is known to be increased in epilepsy and several studies have shown increased CRP levels in the blood. Others have, however, failed to show this increase, which would be in line with our results. We don’t know the reason for these discrepancies. There are several possibilities (e.g. underlying pathology, years of epilepsy, seizure frequency), but this would most likely require the use of more patient samples. We have further highlighted this in our discussion (Line 453):

“An unanticipated finding was, therefore, that CPR levels were similar between TLE patients and controls in our study. However, while a significant association between CRP levels and epilepsy has been observed in some studies [55, 56], this was not confirmed by others in line with our results [17, 57]. The reasons for these discrepancies remain to be determined, however, difference in underlying pathologies and etiologies may explain some of these. This is an important issue, consequently, future studies must be designed to establish why some patients show altered CRP levels while others do not.”

We also agree with the reviewer regarding the correlation of P2X7R with CRP and differences of both biomarkers with respect to control levels. One would expect, if two biomarker levels positively correlate, both should be different from controls. However, while we ensured that the same samples are taken for both studies, unfortunately, not all samples used for our P2X7 studies could be used for our studies analyzing CRP plasma levels, which might be one of the reasons not seeing the same increases. This has been highlighted in the result part (Line 302):

“CRP levels were measured in a subset of samples from the same healthy controls and patients used to quantify P2X7R levels.”

and in the discussion (Line 464):

“Of note, despite the strong correlation of P2X7R and CRP plasma levels 1 h post-seizures, only P2X7R plasma levels, and not CRP plasma levels, were higher when compared to control levels. This is unexpected, as one would expect both, P2X7R and CRP showing the same increases in relation to controls. However, CRP plasma concentrations were only measured in a subset of controls and patients and both CRP and P2X7R plasma levels show a high inter-sample variability which may explain why P2X7R levels but not CRP levels, despite showing a positive correlation 1 h post-seizures, are increased when compared to control.”

  1. Line 29: Abbreviations „TLE” and “PNES” should be explained also in Abstract

This information has been added to the abstract.

  1. Line 109: Since at the mRNA level, P2X7 shows a time of day dependent variation (Lommen  et al, 2017, Cell and Tissue Research 369:579-590), it should be specified at what time exactly the samples were collected.

We agree with the reviewer that this is an important point, in particular for biomarkers which have to be taken at different times of the day (e.g. after an epileptic seizure). However, when comparing P2X7R plasma levels of patients with TLE (baseline) which were taken during the morning (am) (N = 8) with plasma samples taken during the afternoon/night (pm) (N = 8) no differences were found in P2X7R plasma levels between the two groups (p = 0.604)).

This has been added to our Result section (Line: 274):

“Finally, there is evidence that expression of the P2X7R may vary according to time of day [43]. Accordingly, we compared P2X7R levels in a subset of TLE patients. Analysis of plasma levels of P2X7R from TLE patient samples taken in the morning versus afternoon showed no difference (N = 8 per time-point (TLE baseline), p = 0.604).”

We have also added a short paragraph in the Discussion describing the need for future analysis of potential changes of P2X7R plasma levels according to the time of day (Line 496):

“Moreover, P2X7R expression has been reported to change according to the time of day [43]. Future studies should, therefore, investigate whether P2X7R expression in plasma undergoes the same changes.”

  1. Line 246 - 248: “P2X7R plasma levels were higher in baseline samples from patients with TLE (242.6 ± 39.2 pg/ml) compared to controls and levels remained similarly elevated in samples collected 1 h following an EEG-detected seizure (302.1 ± 86.3 pg/ml) (Figure1B)“ But Fig. 1B shows statistical comparison of SE with PNES, not with control, as mentioned in text to this figure: “P2X7R protein levels were also higher in patients suffering from an episode of status epilepticus (SE) (N = 6) when compared to PNES patients” Was the difference between control and TLE statistically significant?  

We would like to apologize for this mistake. The reviewer is right that P2X7R plasma levels were not statistically different between control and patients with SE, which is probably due to the low N number for our SE patient cohort. This has been amended in the text (Line 252).

“While increased by approximately 20% in patients suffering from status epilepticus (287.5 ± 79.7 pg/ml) when compared to control, this did not reach statistical significance (p = 0.0831), which was probably due to the low N number of our status epilepticus patient cohort. P2X7R protein levels were, however, higher in patients with SE when compared to patients with PNES (Figure 1B).”

We also emphasized in the Discussion the need for a bigger patient cohort, in particular regarding our SE patient cohort (Line 492).

“Our results should, however, be further validated in a multi-center study with a larger patient cohort including non-TLE epilepsy patients and patients with status epilepticus to allow for a better understanding of the relationship between plasma P2X7R levels and epilepsy syndromes, underlying disease phenotype and preictal, ictal, postictal and interictal states.”

P2X7R plasma levels are significantly different between Controls and patients with TLE.

  1. Line 308: Correlation

This has been corrected.

  1. Line 327: Number of mice in SE group should be also given

Thank you for pointing this out. We have used 9 mice for each group. This has been added accordingly (Line…):

“(B) Graph showing the histogram of the GFP-A (area) in control conditions and 1 h post-SE. The percentage of GFP+ cells is significantly increased in animals treated with intra-amygdala KA 1 h post-lorazepam (SE) when compared to vehicle-injected control mice (Crtl) (N = 9 per group)”

  1. Lines 450-451: “P2X7R levels seemed to be even below control levels in patients with PNES..“. This differnce was not mentioned to be significant in Results. Moreover, (line 245) the authors stated that " P2X7R levels were similar between patients with PNES and controls (Figure 1B)"

We fully agree with the reviewer. While P2X7R plasma levels seem to be lower in patients with PNES when compared to Controls, this does not reach significance (p = 0.2402). We have, therefore, decided to remove this sentence from the discussion.

  1. Lines 458 -459: “Other sources are, however, also possible including the brain.” This sentence is vague. The authors should specify cells in the brain which could be considered as a P2X7R source - microglia, ependymal cells or astrocytes?

This has been further elaborated in the discussion (Line 480):

“Other sources are, however, also possible including the brain. Among brain cells, microglia are probably the most likely source of increased P2X7R levels in plasma. Microglia are one of the first cell types to respond to injury [60] and P2X7Rs are highly expressed on microglia following status epilepticus and during epilepsy [35]. While not detected on astrocytes, P2X7Rs have, however, also been shown to be increased on neurons and oligodendrocytes during epilepsy [25, 35], making these cell types also possible sources.”

  1. Lines 502 – 511:” Potential explanations for these differences….“. This part of Discussion is full of speculations and poor arguments. For example, what type of P2 receptors, except P2X7, could facilitate release of cytokines? How different could be „different transgenic P2X7-/- mice“?

This part has been extended as suggested by the reviewer in our revised manuscript (Line 532):

“There are several potential explanations for these differences including the use of different transgenic P2X7-/- mice (e.g. P2X7-/- mice previously used to identify P2X7R-dependet cytokines in the blood [74] express several P2X7R splice variants within the CNS [75]), different disease backgrounds or models used (e.g. cells vs whole organism). Other cytokine release-regulating proteins, including other P2 receptors, may become activated during seizures/epilepsy masking the effects of the P2X7R. P2Y1R, P2Y12R and P2X4R have all been shown to be activated during seizures [49] and to regulate the release of cytokines [76, 77].”

Reviewer 2 Report

The manuscript by Conte et al. investigates the expression and functions of the P2X7 receptor in peripheral blood in the context of both human and mouse epileptic brain. The main goal is to assess whether P2X7 can represent a potential biomarker of epilepsy in a clinical context.

Main results are that P2X7 receptor expression is increased in blood cells in both human and mouse following crisis of status epilepticus, but not in patients with psychogenic non-epileptic seizures. Peripheral cytokine profiling in WT and P2X7-deficient mice show that P2X7 negatively control KC/GRO post SE, while other cytokine are not affected. Using P2X7-GFP transgenic mice, monocytes are identified as the main cell population expressing P2X7 post SE. It is concluded that P2X7 signaling component may represent a potential circulating biomarker for epilepsy diagnosis.

The involvement of P2X7 in epileptogenesis and recurrent epileptic crisis has already be demonstrated, although there are some discrepancies in the literature regarding its real contribution. In any case, its contribution to peripheral inflammation has never been investigated, nor its expression in epileptic patients. For these reasons, this study provides clear novel data.

Overall the study is well conducted, the objectives are clearly defined, and results and discussion are fair.

I don’t have any specific comments on the result, methods and discussion. The author should however provide more information regarding the transgenic lines they used. First it is not mentioned how p2x7 was floxed out from the P2rx7tm1d(EUKOMM)wtsi. If I’m not mistaken, this strains is a flox allele that needs the action of a Cre to actually delete the critical exon. This is not detailed in the method section. The author should also discuss the fact that only the second exon is deleted leaving the alternative first exon that is expressed in a subset of T lymphocyte. This precision is important since the study investigate peripheral immune response where the contribution of T lymphocyte cannot be excluded.

Finally, the specific P2X7-GFP line that has been used should be given since in the original paper describing this P2X7-GFP mice, several line were generated, each with different number of transgene insertion.

There are numerous hyphen that are inappropriate, including one in the title.

Author Response

  1. The author should however provide more information regarding the transgenic lines they used. First it is not mentioned how p2x7 was floxed out from the P2rx7tm1d(EUKOMM)wtsi. If I’m not mistaken, this strains is a flox allele that needs the action of a Cre to actually delete the critical exon. This is not detailed in the method section.

According to the reviewer’s recommendations, we have included a more detailed description of our P2X7 mouse line used. This information has been added to the Material and Method section (Line 148):

For our studies we have used Line 17 from the initial generation of P2X7-EGFP mice. This line showed the highest expression of P2X7R-EGFP and has been thoroughly characterized previously [21, 35]. The original B6-P2rx7tm1a(EUCOMM)Wtsi strain was obtained from EMMA and crossed with the FLPe deleter mouse Gt(ROSA)26Sortm1(FLP1)Dym [36] to obtain the P2rx7fl/fl mouse, B6-P2rx7tm1c(EUCOMM)Wtsi. After removal of the FLPe, the P2rx7fl/fl mouse was crossed with the EIIa-Cre mouse, Tg(EIIa-cre)C5379Lmgd [37] to obtain, after removal of the EIIa-Cre, the P2X7-/- mouse. For further details see [21, 38].”

  1. The author should also discuss the fact that only the second exon is deleted leaving the alternative first exon that is expressed in a subset of T lymphocyte. This precision is important since the study investigate peripheral immune response where the contribution of T lymphocyte cannot be excluded.

P2X7 receptors depend on the presence of both transmembrane domains, to
form functional receptors. Since TM1 is partly encoded by exon 2,
deletion of this exon should result in the knock-out of all functional
rodent splice variants, including P2X7a and P2X7k. Nevertheless, we have added a short paragraph to the discussion bringing this issue up (Line 544):

“One potential concern with the use of the current P2X7-/- mouse model is the inclusion of exon 1. This could potentially leave functional P2X in lymphocytes. However, P2X7Rs depend on the presence of both transmembrane domains (TMs) to form functional receptors. Since TM1 is partly encoded by exon 2, deletion of this exon should result in the knock-out of all functional rodent splice variants, including P2X7a and P2X7k.”

  1. Finally, the specific P2X7-GFP line that has been used should be given since in the original paper describing this P2X7-GFP mice, several line were generated, each with different number of transgene insertion.

This has been added to the Materials and Methods section (see answer under point 1).

  1. There are numerous hyphen that are inappropriate, including one in the title.

This has been corrected accordingly.